# IDH Inhibitors and Immunotherapy for Biliary Tract Cancer: A Marriage of Convenience?

**DOI:** 10.3390/ijms231810869

**Published:** 2022-09-17

**Authors:** Giovanni Brandi, Alessandro Rizzo

**Affiliations:** 1Department of Specialized, Experimental and Diagnostic Medicine, University of Bologna, Via Giuseppe Massarenti, 9, 40138 Bologna, Italy; 2Division of Medical Oncology, IRCCS Azienda Ospedaliero-Universitaria di Bologna, Via Albertoni, 15, 40138 Bologna, Italy; 3Struttura Semplice Dipartimentale di Oncologia Medica per la Presa in Carico Globale del Paziente Oncologico “Don Tonino Bello”, I.R.C.C.S. Istituto Tumori “Giovanni Paolo II”, Viale Orazio Flacco 65, 70124 Bari, Italy

**Keywords:** biliary tract cancer, immunotherapy, immune checkpoint inhibitors, cholangiocarcinoma, durvalumab, ivosidenib

## Abstract

Systemic treatments have traditionally reported limited efficacy for biliary tract cancer (BTC), and although targeted therapies and immune checkpoint inhibitors have been found to play an increasingly important role in treatment, several questions remain unanswered, including the identification of biomarkers of response. The tumor microenvironment (TME) has recently attracted the attention of the BTC medical community, and is currently being studied due to its potential role in modulating response and resistance to systemic therapies, including immunotherapy. In this perspective article, we discuss available evidence regarding the interplay between TME, IDH inhibitors, and immunotherapy, providing rationale for the design of future clinical trials.

## 1. Introduction

Biliary tract cancer (BTC) includes a heterogeneous group of aggressive and rare gastrointestinal (GI) malignancies, including extrahepatic cholangiocarcinoma (eCCA), which is classified into perihilar (pCCA) and distal cholangiocarcinoma (dCCA), intrahepatic cholangiocarcinoma (iCCA), and gallbladder cancer (GBC) [1,2]. BTC represents the second most frequent primary liver tumor following hepatocellular carcinoma (HCC), accounting for approximately 10–15% of all primary liver cancers and 3% of all GI malignancies [3]. When surgically feasible and when diagnosed at an early stage, radical surgery is the standard-of-care treatment for BTC [4]; however, potentially curative surgery is possible only in a minority of BTCs, since most patients present with unresectable metastatic disease, where systemic chemotherapy is the standard therapeutic approach. The combination of gemcitabine plus cisplatin (GemCis) has represented the first-line standard for the treatment of advanced BTC for more than a decade, based on the results of the practice-changing ABC-02 study showing a median overall survival (OS) of around one year [5,6]; of note, the recently presented and published results of the TOPAZ-1 phase III trial highlighted prolonged survival for BTC patients receiving a combination of the PD-L1 inhibitor durvalumab plus GemCis versus GemCis alone, with this evidence supporting a novel first-line standard [7]. In addition, pemigatinib and infigratinib in the molecular subset of fibroblast growth factor receptor-2 (FGFR2)-rearranged tumors and ivosidenib in isocitrate dehydrogenase 1 (IDH1)-mutant tumors have become FDA-approved options for previously treated advanced BTC harboring targetable alterations, opening the doors of a new era in this setting [8,9,10].

A key point to consider regarding the role of chemoimmunotherapy in BTC is certainly the lack of validated biomarkers, with TOPAZ-1 reporting clinical benefit in an unselected population of patients with these hepatobiliary malignancies. In recent years, we have seen a growing interest towards a deeper understanding of BTC tumor biology, and several molecular features have been identified. Among these, BTCs have been suggested to present a desmoplastic tumor microenvironment (TME), being characterized by a high heterogeneity, as observed by several reports [11]. In addition, the TME is currently being studied due to its potential role in modifying the response to anticancer treatments, including immune checkpoint inhibitors and targeted therapies [12]. In this perspective article, we discuss available evidence regarding the interaction between TME, IDH inhibitors and immunotherapy in BTC, providing rationale for the design of future clinical trials. 

## 2. IDH Inhibitors and Tumor Microenvironment in Biliary Tract Cancer

As previously stated, IDH1 and FGFR2 are considered crucial therapeutic targets in BTC harboring specific genetic aberrations, and practice-changing clinical trials have been recently presented and published, including the ClarIDHy phase III study [13,14]. At the same time, preclinical evidence has highlighted that specific genomic aberrations may alter the complex mechanisms of interaction between BTC, the immune system, and immunotherapeutic agents. In particular, although IDH1 inhibitors such as ivosidenib have provided clinical benefit for patients with IDH1 mutations, response rates are low. For example, the disease control rate (DCR) observed with ivosidenib in ClarIDHy was due to stable disease (51%), with only 2% showing partial responses [13,14].

Based on these premises, some recent studies have suggested that mutations in IDH1 may modify the tumor immune landscape, and some approaches oriented to use these mutations as therapeutic targets are being tested. From a biological point of view, mutations in IDH1 and IDH2 genes have been detected in several hematological and solid tumors, such as glioblastomas, chondrosarcomas, acute myeloid leukemia, thyroid malignancies, myeloproliferative tumors, and cholangiocarcinomas [15,16,17]. For example, mounting evidence has elucidated the biological impact of IDH mutations and has also uncovered the clinically relevant role of these aberrations, with the development of novel treatments able to target IDH-mutated tumors. These mutations induce a gain-of-function activity, which leads to the conversion of α-ketoglutarate (α-KG) to the oncometabolite D-2-hydroxyglutarate (2-HG) (Figure 1) [18]; the latter accumulates in cancer cells and is secreted at high levels into the TME, with 2-HG being able to inhibit several enzymes which use α-KG for their activity, including regulators of crucial processes such as cell differentiation and cell metabolism. Elevated 2-HG levels severely impair cellular metabolism, increase reactive oxygen species, cause DNA hypermethylation, and contribute to oncogenesis, even in synchronous or metachronous different tumors (e.g., glioblastomas and cholangiocarcinomas) [19]. Thus, there is growing evidence supporting the presence of mechanisms involved in the modulation of immune phenotypes of IDH1-mutated malignancies. For example, a pan-cancer analysis of bulk transcriptomics data reported that IDH1 mutations may be associated with gene signatures indicative of low B lymphocytes, NK cells, and T lymphocyte infiltration [20]. Regarding iCCAs, some transcriptomics analyses have shown an association between IDH mutations and low intra-tumoral CD8+ T cells, as well as lower immune-related signaling compared with IDH1 wild-type tumors. In another solid tumor commonly reporting IDH1 mutations, flow cytometric, immunohistochemical, and single-cell RNA sequencing analyses have observed that tumors harboring IDH1 mutations may be considered immunologically “cold” compared to wild-type gliomas, given the lower presence of CD8+ T cells and Treg cells [21,22]. 

Preclinical evidence suggests that mutations in IDH1 may be involved in immunosuppressive signals, and thus, IDH activity seems to play a key role in modulating the activity of the immune system [23]. In a recent study published by Wu and colleagues, the authors observed a fast recruitment of infiltrating CD8+ T cells in a genetically engineered murine model of IDH1-mutated iCCA, resulting in a remarkable reduction in tumoral growth [24]. At the same time, Wu et al. reported that the depletion of CD8+ T cells inhibited the anticancer activity of the IDH1 inhibitor, something that poses some questions regarding the role of immune evasion as a fundamental process modifying the action of ivosidenib and other anti-IDH agents.

## 3. Immune-Based Combinations including IDH Inhibitors: The Next Frontier?

Preclinical studies have evidenced that inhibiting IDH1 may stimulate anticancer immunity through the conversion of an immunologically “cold” milieu to “hot”, as well as by determining the restoration of tumor cells’ sensitivity to immunological signals (Figure 2). In the previously cited study by Wu and colleagues, treatment with ivosidenib provoked the recruitment of CD8+ T cells, which in turn stimulated the production of interferon-gamma, causing the upregulation of antigen presentation, proliferative arrest, and finally, cell death [24]. From a mechanistic point of view, ivosidenib causes the de-repression of a target of 2-HG, the TET2 demethylase, in cancer cells, which in turn plays a crucial role in the promoter demethylation and the epigenetic upregulation of target genes related to interferon-gamma. Of note, the deletion of either TET2 or the interferon-gamma-receptor 1 (IFNGR1) has been associated with the resistance to IDH1 inhibition, despite neither TET2 nor IFNGR1 deletion preventing the recruitment of CD8+ T cells [24]. Therefore, the TET2 pathway has an important role in modifying the response to ivosidenib in tumors harboring IDH1 mutations. 

Available evidence suggests that mutations in IDH1 may favor immunosuppression through several mechanisms; among these, 2-HG may be secreted and be detected directly in the TME, as well as in the urine and plasma of IDH-mutated cancer patients. Based on these premises, some studies have also reported paracrine effects determined by secreted 2-HG on immune cell types such as macrophages and T cells [25]. 2-HG can impair mitochondrial respiration in human CD4+ and CD8+ T cells, and thus, may alter the proliferation and production of interferon-gamma and IL-2. In glioma models, 2-HG has been also suggested to enhance the tryptophan degradation pathway in myeloid cells to promote reprogramming to immunosuppressive macrophages. However, few data are currently available and further evidence is needed to define the players involved in CD8+ T cell recruitment and the immune effects determined by these genetic aberrations. Another interesting research avenue regards how IDH inhibitor treatment may modify the function of some molecular pathways, such as JAK/STAT3, with several studies reporting that STAT3 levels in the TME may be associated with primary and secondary resistance to ICIs in solid tumors, including non-small cell lung cancer [26,27]. Moreover, the near future will better tell us how one of the most important types of stromal cells—cancer-associated fibroblasts (CAFs)—with a fundamental role in the TME may interact with BTC cells, immune cells, and anticancer treatments such as immunotherapy and IDH-targeted agents.

In our opinion, restoring antitumor immunity by inhibiting IDH1 mutations provides the basis for testing combinatorial strategies based on immune checkpoint inhibitors or chemoimmunotherapy plus IDH inhibitors [28]. For example, some reports have suggested that disease progression in iCCAs receiving the IDH1 inhibitor ivosidenib was associated with the induction of PD-L1 in cancer cells, Treg recruitment, and higher PD-1, CTLA-4, and CD80 expression in the TME [29,30]. At the same time, the use of a CTLA-4 antibody was reported to increase ivosidenib efficacy. In summary, a strong biological rationale supports the testing of immune-based combinations, including IDH1 inhibitors. Since the recently published results of the TOPAZ-1 trial may lead to a new era in BTC management, the design of future clinical trials on first-line therapy for patients harboring IDH mutations should be focused on this topic and the possibility to add anti-IDH agents to chemoimmunotherapy. We are hopeful that ongoing clinical trials will shed light on these burning questions and expand the therapeutic opportunities in this dismal malignancy with—still—many unanswered questions.

## Figures and Tables

**Figure 1 ijms-23-10869-f001:**
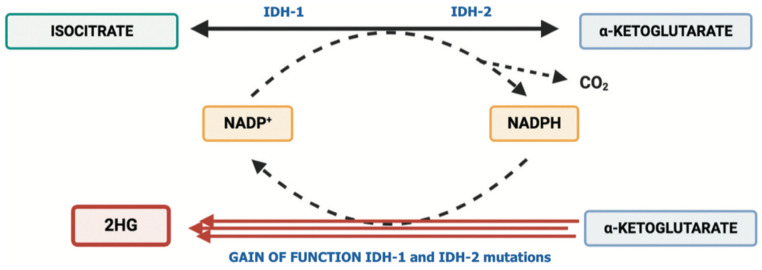
Schematic representation of IDH1 and IDH2 mutations determining the pathological accumulation of 2-HG.

**Figure 2 ijms-23-10869-f002:**
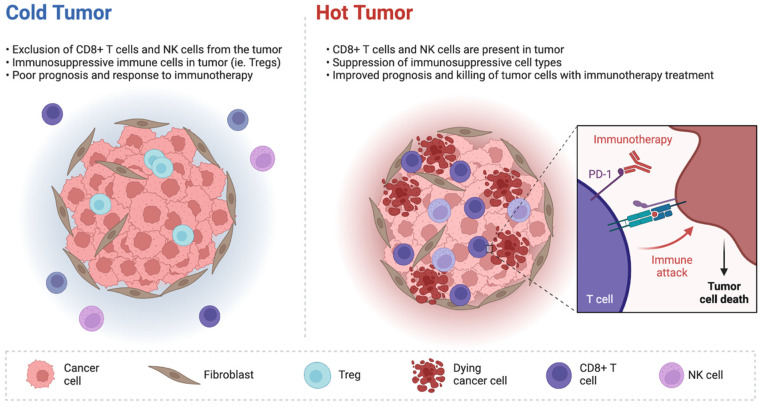
Schematic representation of immunologically “cold” and “hot” tumors.

## Data Availability

Not applicable.

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
