# Peer review of "IDH Inhibitors and Immunotherapy for Biliary Tract Cancer: A Marriage of Convenience?"

_ijms, 2022, doi:10.3390/ijms231810869_

Round 1

Reviewer 1 Report

In this review authors discuss the evidences regarding the interaction between IDH inhibitors, tumour microenvironment and immunotherapy in biliary tract cancer. They suggest that restoring antitumor immunity by inhibiting IDH1 mutations could suppose novel strategies to treat biliary tract cancer patients.

MINNOR COMMENTS:

-        Could authors include some additional information regarding other mechanisms of resistance to immunotherapy that could be modulated after IDH inhibitors treatment? i.e.: Some pathways that could be altered after IDH inhibitors treatments (such as JAK/STAT3) and that could also affect to immunotherapy response (STAT3 levels in tumour microenvironment have been related with immunotherapy resistance in some preclinical models of lung cancer)

-        CAFs (Cancer-Associated Fibroblasts) and TAMs (Tumor-Associated Macrophages) are key cells in TME and have been described as important cells that could alter the immunotherapy resistance of some tumours including cholangiocarcinoma. Could authors include some information about the role of these cells in TME after ivosidenib treatment?

Author Response

Dear Reviewer, 

Thank you so much for your comments.

  1. We better discussed and reported that IDH inhibitors may modify the activity of some molecular pathways, such as JAK/STAT3, and we included two very important reports regarding this topic.
  2. Thank you for this suggestion. We briefly discussed this emerging and promising topic, as suggested.

All our changes have been reported in red color. 

Thank you again for your comments. We hope the revised paper will better suit the journal.

Reviewer 2 Report

The perspective written by Rizzo et al, entitled “IDH inhibitors and immunotherapy for biliary tract cancer: a marriage of convenience?” elaborated the use IDH1 inhibitor in mutated biliary tract cancer (BTC) patients. They also suggested the border view on use of IDH1 inhibitor in combination of chemotherapy and/or PD-L1 blockade and immunotherapy,

The manuscript was well written,

Limitation:

1)      Overall, the script has very lesser view on the broad field, on their extensive studies of IDH mutation since their discovery from 2008 on glioblastoma and AML

Author Response

Dear Reviewer,

Thank you for the time spent revising our paper.

We better discussed this topic, as suggested.

Our changes have been reported in blue and red.

We hope the revised manuscript will better suit the journal.